# OpenReview forum: "Online Detecting LLM-Generated Texts via Sequential Hypothesis Testing by Betting"
_ICLR.cc/2025/Conference — Submitted to ICLR 2025_

### Official Review · Reviewer_NdvJ · 2024-10-29

**Soundness:** 3
**Presentation:** 2
**Contribution:** 3
**Rating:** 5
**Confidence:** 3

**Summary:**

This paper proposes to detect LLM-generated texts in an online streaming scenario, addressing the gap in current detection methods that primarily operate offline. The authors introduce an algorithm based on sequential hypothesis testing through a betting mechanism, allowing for real-time, statistically robust detection with controlled false-positive rates and bounded detection times. Experimental results demonstrate that the method can effectively distinguish LLM-generated text sequences from human-written texts, even under practical constraints.

**Strengths:**

* The paper explores the detection of generated text in a streaming context, which is timely and might be useful for real-world applications such as monitoring social media or news platforms.
* The authors build on the theory of sequential hypothesis testing by betting, establishing clear statistical guarantees. This foundation provides rigor, including control over type-I errors and expected detection times, which is valuable for applications requiring reliable, real-time decisions.
* The authors test their approach using different scoring models, source models, and baselines, showing a commendable effort in evaluating robustness and sensitivity to various parameters.

**Weaknesses:**

* The scope of the paper’s application could be better defined. For example, is the goal to identify whether the content in a stream is entirely machine-generated or to ascertain the overall nature of the source (e.g., a bot account)? If the latter, the assumption that all content from a source is generated by a single LLM may not hold in practice, where mixed content (mixed LLMs or mixed human and LLM-generated text) is common. How the method would handle mixed content or bot accounts using various LLMs is unclear.
* The experiments rely on a single dataset and use a relatively small amount of data, which weakens the persuasiveness of the results. Furthermore, the generated texts tested in this study are produced by limited models; the evaluation would be more compelling if it included results from more advanced models, such as LLaMA 3, Mistral, or GPT-4.
* The writing can be polished for clarity and completeness. Many of the experimental result tables are placed in the appendix, which could have been included in the main text given there is remaining space.

**Questions:**

* How does the method handle situations where the source content may contain a mixture of human-written and machine-generated text?
* Could the sequential hypothesis testing technique adapt dynamically to model drift or evolution in LLMs over time?

---

> ### Author Response · Authors · 2024-11-18
> **Authors' response (1/2)**
>
> We thank the reviewer for your constructive feedback. We value your feedback and look forward to discussing it further.
>
> &nbsp;
>
> * __The scope of the paper’s application could be better defined. For example, is the goal to identify whether the content in a stream is entirely machine-generated or to ascertain the overall nature of the source (e.g., a bot account)? If the latter, the assumption that all content from a source is generated by a single LLM may not hold in practice, where mixed content (mixed LLMs or mixed human and LLM-generated text) is common. How the method would handle mixed content or bot accounts using various LLMs is unclear.__
>
> Thanks for the feedback. While we mentioned that we concern the scenario where the texts from the unknown source are all generated either by human or by an LLM, we note that it is trivial to extend our method to the settings that the reviewer is proposing. Specifically, when texts from the unknown source are produced by various LLMs, the alternative hypothesis would still hold, which implies that our statistical guarantees remain valid, and the algorithm could continue to perform effectively.
> To support this claim, we have conducted a new experiment and include the result in the updated version. Please kindly find Figure 6 (a) in page 10 of our updated manuscript for the result of this scenario.
>
> Furthermore, when the unknown source publishes both human-written and LLM-generated texts, our method can effectively tackle this setting as well. Here, the null hypothesis assumes that all texts from the unknown source are human-written. In contrast, the alternative hypothesis posits that not all texts are human-written, which indicates the presence of texts generated by LLMs.
> We have added a new experiment for the scenario, which can be found
> in Figure 6 (b).  The results demonstrate that our algorithm, equipped with nearly all score functions, consistently performs well in this new setup.
>
> We appreciate the reviewer for this feedback, which helps strengthen the presentation and results of our work.
>
> &nbsp;
>
> * __The experiments rely on a single dataset and use a relatively small amount of data, which weakens the persuasiveness of the results. Furthermore, the generated texts tested in this study are produced by limited models; the evaluation would be more compelling if it included results from more advanced models, such as LLaMA 3, Mistral, or GPT-4.__
>
> We would like to clarify that additional experimental results for online testing are provided in Appendix H of our paper.
> In these experiments, samples of LLM-generated texts and human-written texts (from datasets such as XSum, WritingPrompts, and PubMed) are from existing sources in the literature [1], with the LLM-generated texts produced by models such as GPT-3, ChatGPT, and GPT-4. We also believe that Gemini, which we used to construct the new dataset, is one of the most advanced models.
> Our experiments did not rely on a single dataset or use a relatively small amount of data.
>
> &nbsp;
>
> * __The writing can be polished for clarity and completeness. Many of the experimental result tables are placed in the appendix, which could have been included in the main text given there is remaining space.__
>
> We thank the reviewer's feedback regarding the placement of our experimental result tables. However, we only have less than one page (of Page 10) remaining in the main text, which does not allow for the inclusion of multiple long tables (with each can take one-page long) without breaking the consistency of information. To address this, we have ensured that each table in the appendix is referenced in the main text and is presented in a logical sequence. We also highlight key findings of these tables in the result section in order to ensure readers can grasp the essential insights and thoroughly explain the results in detail in the appendix.
>
> On the other hand, a couple of new experiments have conducted in addressing the reviewer's questions, and we have used Page 10 to report the main parts of the new results.
>
> &nbsp;

---

> ### Author Response · Authors · 2024-11-18
> **Authors' response (2/2)**
>
> * __Could the sequential hypothesis testing technique adapt dynamically to model drift or evolution in LLMs over time?__
>
> Yes, we believe that sequential hypothesis testing can handle the case of the model drift and the case of evolution in LLMs over time. Although it is outside the scope of our work, we briefly describe a potential approach to tackle model drift/shift here. We denote the population distribution of the articles at time $t$ by $h_t$, and the model at that time by $\phi_t$. To establish the hypotheses in this setting, we can define $\Delta_t$ as the absolute difference $|\mu_x(t) - \mu_y(t)|$, where $\mu_x(t) = E_{X\sim h_t} [\phi_t(X)| F_{t-1}]$, where $F_{t-1}$ is the filtration up to and including $t-1$. This could lead us to formulate the null hypothesis and the alternative hypothesis as follows:
>
> $$
> H_0 : \Delta_t = 0 \quad \text{for all } t \geq 1, \\\
> H_1 : \exists T \in \mathbb{N} \text{ such that } \Delta_t > 0 \quad \text{for all } t \geq T.
> $$
> Besides, we would like to note that our new experiment of the sequence of texts coming from various LLMs as the reviewer suggested could also be viewed as a model drift problem. This is because the mean of the text scores from different LLMs are different, i.e., $\mu_y(t)$ is not a constant in this scenario. Hence, our experimental result suggests that handling the model drift or distribution shift is a feasible direction. We thank the reviewer again for this feedback!
> &nbsp;
>
> [1] Guangsheng Bao, Yanbin Zhao, Zhiyang Teng, Linyi Yang, and Yue Zhang. Fast-detectgpt: Efficient zero-shot detection of machine-generated text via conditional probability curvature. arXiv preprint arXiv:2310.05130, 2023.

---

> > ### Comment · Reviewer_NdvJ · 2024-11-24
> >
> > I appreciate the author's response and I have read all the reviewers' comments. I tend to keep the current score.

---

> > > ### Author Response · Authors · 2024-11-24
> > >
> > > We thank the reviewer for your reply. We would greatly appreciate it if the reviewer could consider upgrading the score if our response and updates to the paper have sufficiently addressed the reviewer's comments.

---

### Official Review · Reviewer_87z9 · 2024-11-04

**Soundness:** 2
**Presentation:** 3
**Contribution:** 2
**Rating:** 3
**Confidence:** 2

**Summary:**

This paper proposes a novel method for the online detection of text generated by large language models (LLMs), employing sequential hypothesis testing based on betting techniques. The approach aims to provide robust statistical guarantees, balancing type-I error control with efficient detection times. By leveraging sequential hypothesis testing, the method ensures accurate identification of LLM-generated text while minimizing false positives.

**Strengths:**

1. The application of sequential hypothesis testing with betting is innovative and well-suited to the online detection problem, distinguishing this approach from conventional offline methods.
2. The method includes strict statistical guarantees for false positive rates and expected detection times, which is critical for applications requiring high reliability.
3. The paper demonstrates solid technical grounding, and a wealth of theoretical knowledge, and is well-written.

**Weaknesses:**

1. The abstract lacks clarity, particularly in the phrasing "publish content in a streaming fashion." The motivation should emphasize the need for continuous online detection due to the constant emergence of text, which differs from offline detection. This ongoing process allows for enhanced detection performance through hypothesis testing.
2. Based on this motivation, the baseline experimental setup appears problematic. It should compare your method with current offline approaches (e.g., detection without leveraging continuous information) rather than with other hypothesis tests, to better evaluate effectiveness and identify advantages and disadvantages.
3. The scoring model has not been optimized for this task. While experiments on text domain matching were conducted, no solutions were proposed. The experiments are somewhat limited; the focus should be on the modules and parameter settings of sequential hypothesis testing compared to offline methods. Ablation studies should emphasize this point.

**Questions:**

1. Why introduce a new dataset? Are the existing datasets inadequate? Is 500 samples insufficient for credibility? Why generate only one new dataset instead of two or three? How can we ensure that the series of input texts y_x come from the same source?
2. The examples provided at the beginning are unconvincing; they primarily address account and regulatory issues. Why not present more compelling examples that better reflect your motivation for the revision?
3. For hyperparameters such as d_* and epsilon, is there an optimal selection method? What are the trade-offs involved?

---

> ### Author Response · Authors · 2024-11-18
> **Authors' response (1/2)**
>
> Thank you for your comments and questions. We would like to provide further clarification on the following points:
>
> &nbsp;
>
> * __Based on this motivation, the baseline experimental setup appears problematic. It should compare your method with current offline approaches (e.g., detection without leveraging continuous information) rather than with other hypothesis tests, to better evaluate effectiveness and identify advantages and disadvantages.__
>
> We want to emphasize that the online detection task in our work is about obtaining an anytime-valid $\alpha$ level test (i.e., controlling the false positive rate when the source is human) while controlling the time to reject the null hypothesis when the alternative is true (i.e., controlling the time to correctly identify the source as an LLM), where a sample from the unknown source arrives sequentially.
> The ``anytime-valid'' here means that the guarantees hold at any stopping time.
> To the best of our knowledge, this is a novel online detection scenario of detecting LLMs, and therefore existing works for offline classification cannot be directly applied to our online scenario. For example, a naive way to adapt the offline detector in our scenario is that if at a certain point, the offline detector predicts that a sample is written by LLM, then it declares the source is LLM (and hence stops).  However, this naive method will have a false positive rate of 1 when the number of rounds becomes sufficiently large, unless the offline detector never makes an error in recognizing human text.
>
> &nbsp;
>
> * __The scoring model has not been optimized for this task. While experiments on text domain matching were conducted, no solutions were proposed. The experiments are somewhat limited; the focus should be on the modules and parameter settings of sequential hypothesis testing compared to offline methods. Ablation studies should emphasize this point.__
>
> In our experiment, two scoring models used are both very effective to get low false positive rate (FPR) and short rejection time. Furthermore, most of the score functions perform well, and Fast-DetectGPT shows the best performance with the scoring model Neo-2.7. The experimental results verify the effectiveness of the statistical guarantees.
>
> As mentioned previously, we focus on a novel scenario which is different from that of the offline detection. Our method is based on the existing offline detectors and complements them, not a competitive method to them. Also, we believe the ablation studies is not applicable since our approach builds upon those score functions (the score functions are essential and cannot be removed).
>
> &nbsp;
>
> * __Why introduce a new dataset? Are the existing datasets inadequate? Is 500 samples insufficient for credibility? Why generate only one new dataset instead of two or three? How can we ensure that the series of input texts $y_x$ come from the same source?__
>
> We first would like to emphasize that we also conducted experiments using existing datasets of [1], and the results are available in Appendix H due to page limitations. The reason why we introduced the new dataset is because we want to further validate the effectiveness of our approach in real-world scenarios. In our experiments, we found that with most configurations, our algorithms can detect LLM-generated texts before the time budget 500 (e.g., Figure 3).
>
> As detailed in the Experiment Settings section and reiterated in the figure captions, $x_t$ represents text sampled from a prepared human text dataset, and $y_t$ denotes the text whose source we aim to detect. We recall that the problem setup is that under the null hypothesis, the sequence of $y_t$ are human-written texts, while under the alternative hypothesis, $y_t$ is generated by LLMs. For the latter case, we recorded the rejection time, which is the number of time steps it took to determine that the texts were LLM-generated. We want the rejection time to be as small as possible to quickly declare the source is LLM when it is.
>
> We emphasize the the setting can be trivially extended to the
> scenario that the unknown source publishes both human-written and LLM-generated texts. That is, our method can effectively tackle this setting as well. Here, the null hypothesis assumes that all texts from the unknown source are human-written. In contrast, the alternative hypothesis posits that not all texts are human-written, which indicates the presence of texts generated by LLMs.
> We have added a new experiment for the scenario, which can be found
> in Figure 6 (b) on page 10.  The results demonstrate that our algorithm, equipped with nearly all score functions, consistently performs well in this new setup.

---

> ### Author Response · Authors · 2024-11-18
> **Authors' response (2/2)**
>
> * __The examples provided at the beginning are unconvincing; they primarily address account and regulatory issues. Why not present more compelling examples that better reflect your motivation for the revision?__
>
> We first would like to note that other reviewers do not have such comment. These examples were carefully selected to illustrate not only the applications of machine-generated texts but also the potential misuses of large language models across various public domains, extending beyond account regulation. For instance, the use of GPT-2 to overwhelm public comment systems during policy revisions [2] or GPT-J to manipulate political discussions [3] highlights the urgent need to quickly detect the source of potentially harmful texts posted by users. This underscores the necessity of our method in various online scenarios and strongly aligns with the motivation behind our research to develop online detection algorithms with robust statistical guarantees.
>
> We wish to emphasize that existing offline detection methods generally are for the standard classification setting, while many real-world scenarios involve texts that are sequentially generated, particularly on public websites and social media.  To tackle the challenges of the online detection, we hence consider the online testing scenario that involves maintaining an anytime-valid level test while controlling the time to reject the null hypothesis when the alternative is true (i.e., controlling the time to identify the source as an LLM), where a sample from the unknown source arrives sequentially.
>
> We would also like to note that the problem and the solution we propose are particularly useful when one seeks substantial savings in both data collection and time without compromising the reliability of their statistical testing. These desiderata might be difficult to achieve with approaches based on collecting data in batches (or offline).
>
> &nbsp;
>
> * __For hyperparameters such as $d*$ and $\epsilon$, is there an optimal selection method? What are the trade-offs involved?__
>
> We would like to clarify that the task here is parameter estimation rather than hyperparameter tuning. Therefore, we are not selecting values over a grid. We note that $d_*$ is an upper bound for the differences in text scores $|\psi(x_t) - \phi(y_t)|$, which is used to maintain nonnegative wealth, and $\epsilon$ represents the discrepancy between scores of texts written by different humans. This parameter is motivated by potential variations in human-written text distributions in real-world scenarios.
>
> We also emphasize that if prior knowledge is available, estimating the required parameters is unnecessary. For instance, if one already knows the output range of an off-the-shelf score function and the discrepancy in scores written by different humans, the estimation is not required. This corresponds to the oracle scenario in the experiments, which could be realistic in certain cases.
>
> &nbsp;
>
> [1] Guangsheng Bao, Yanbin Zhao, Zhiyang Teng, Linyi Yang, and Yue Zhang. Fast-detectgpt: Efficient zero-shot detection of machine-generated text via conditional probability curvature. arXiv preprint arXiv:2310.05130, 2023.
>
> [2] Max Weiss. Deepfake bot submissions to federal public comment websites cannot be distinguished from human submissions. Technology Science, 2019121801, 2019.
>
> [3] Leon Frohling and Arkaitz Zubiaga. Feature-based detection of automated language models: tackling gpt-2, gpt-3 and grover. PeerJ Computer Science, 7:e443, 2021.

---

> > ### Author Response · Authors · 2024-11-25
> >
> > Dear Reviewer 87z9,
> >
> > We appreciate the time you've taken to review our manuscript. We provided detailed responses to your concerns a week ago and would like to confirm whether these responses have effectively addressed your concerns.

---

### Official Review · Reviewer_p6Pn · 2024-11-08

**Soundness:** 3
**Presentation:** 3
**Contribution:** 2
**Rating:** 5
**Confidence:** 3

**Summary:**

The authors study the problem of inferring whether a source of texts is human or an LLM in an online setting, i.e, in which texts are observed as a continuous stream, as in social media and online forums.

The paper employs a sequential hypothesis testing framework, allowing the judgment about the likelihood of a source being a LLM to be made dynamically rather than waiting for the entire dataset. The framework is fed with state-of-the-art offline approaches to detect LLM-generated text, and at each step the "wealth" of the system is updated based on the probability of a text being generated by an LLM. This accumulated wealth serves as evidence to reject the null hypothesis (that the text was written by a human) and declare the source as an LLM.

The paper discusses the statistical guarantees of the algorithm, including the controlled false positive rate and the average time to correctly detect an LLM. Additionally, experimental results are presented, demonstrating the effectiveness of the proposed method compared to baseline methods.

**Strengths:**

The paper is well-written paper, easy to understand, the method is defined clearly. The experimental results, considering the absolute numbers,  seem to outperform the employed baselines. The method is intuitive and the weight assignment strategy may help the interpretability of the results. The originality is a bit limited, since it basically adapts sequential hypothesis testing for detecting LLM-generated texts. The significance is still rather limited since they use a dataset that they constructed and it is not published or employed in other works.

**Weaknesses:**

First of all, I believe that authors should better characterize how online detection differs from the offline detection from an algorithmic perspective. I understand that latency and throughput became main requirements, but it is not clear to me, with respect to content handling how it differs from the offline problem, for which there are a large number of works published. Interestingly enough, there are no computational performance assessments in the paper.

The evaluation seems to me as the main weakness to the manuscript. There seems to be several design and experimental decisions that are quite adhoc. For instance, the authors mention that they used 10 score functions in total. How did these functions chosen? Why? Similar issues arise regarding the dataset, which were constructed by the authors and it is not detailed how. For instance, they mention that they generate fake news considering the first 30 tokens of each real news. Is there a validation that the generated fake news are really fake? To what extent?  I also missed more actual baselines that have been employed on the task or similar ones.

Although the authors provided a quite detailed description of the experimental setup, it is not clear to what extent it is reproducible. For instance, in section 4.1 the authors mention that they constructed a dataset containing both real and fake news, but both the construction process and the characteristics of the resulting dataset are not provided.

**Questions:**

I miss a contextualization of the state of the art of LLM-generated text detection – are those algorithms effective?

It seems that an important characteristic of the problem tackled by the authors is that the paper tries to classify sources instead of individual documents – hence, the signal and confidence is dramatically higher than when you have to judge a single document. Consider highlighting this and take a look at https://arxiv.org/html/2406.07016v1.

We do need both validation and characterization of the dataset used in the experiments.

---

> ### Author Response · Authors · 2024-11-18
> **Authors' response (1/2)**
>
> Thanks for your good suggestions. We appreciate the opportunity to clarify the following points.
>
> &nbsp;
>
> * __The significance is still rather limited since they use a dataset that they constructed and it is not published or employed in other works.__
>
> We would like to clarify that there are more experimental results of the online testing in Appendix H of our paper, where samples of LLM-generated texts and human-written texts are from some existing datasets [1].
> We also have provided the code and the new dataset that we construct in the supplementary file for the reader to reproduce our result.
>
> &nbsp;
>
> * __I believe that authors should better characterize how online detection differs from the offline detection from an algorithmic perspective. There are no computational performance assessments in the paper.__
>
> We appreciate the reviewer's feedback here. We consider a novel scenario where texts are observed sequentially with the goal of any-time valid statistical guarantees, while related works (of offline detection) concern classifying a given set of texts with no such guarantees. Specifically, existing methods in offline detection try to determine whether a text is from an LLM by comparing its evaluated text score with a pre-defined threshold. The threshold is very important for these methods, which affects their accuracy, and is always chosen by training on some datasets beforehand. While our method only uses an offline detector to score texts, the threshold/parameter in our algorithm (i.e., the significance level $\alpha$ that a user wants to control) is specified according to the user's need. We have provided the analysis that shows our algorithm can control the false positive rate (below $\alpha$). Our method maintains an any-time valid level-$\alpha$ test while controlling the time to reject the null when the alternative is true as shown in our paper.
>
> &nbsp;
>
> * __Computational performance assessments.__
>
> We consider the novel online setting and concern the rejection time under the alternative hypothesis (i.e., time to identify that the source is an LLM). To the best of our knowledge, there has not been any related work on the online detection task considered in our paper. Moreover, as mentioned in the manuscript, our method is based on offline detectors and complements them, rather than competing with them.
>
> &nbsp;
>
> * __For instance, the authors mention that they used 10 score functions in total. How did these functions chosen? Why?__
>
> There are various score functions in the literature, and hence we can not exhaustively consider all of them. The choice of $10$ score functions was guided by established research in the field (for offline detection); we specifically followed [1], which covers zero-shot detectors, supervised classifiers, and some SOTA detectors like Fast-DetectGPT.
>
> &nbsp;
>
> * __Similar issues arise regarding the dataset, which were constructed by the authors and it is not detailed how. For instance, they mention that they generate fake news considering the first 30 tokens of each real news. Is there a validation that the generated fake news are really fake? To what extent?__
>
> Thanks for providing the feedback. We have provided the code and the new dataset that we construct in the supplementary file for the reader to reproduce our result. We also have updated our paper accordingly to elaborate more on how we construct the dataset.
> Furthermore, we would like to emphasize that there are more experimental results of the online testing in Appendix H of our paper, where samples of LLM-generated texts and human-written texts are from some existing datasets [1].
>
> We follow the steps in [1,2,3] to construct the new dataset, which produces machine-generated texts by prompting with the first 30 tokens from some human-written texts. We use the T5 tokenizer to process each human-written news article to retrieve the first 30 tokens as \{prefix\}. Then, we initiate the generation process by sending the following messages to the model service, such as: "You are a News writer. Please write an article with about $150$ words starting exactly with \{prefix\}." We have included the description of generation process in the updated manuscript, and more details can be found in codes and datasets in our supplementary materials.
>
> &nbsp;
>
> *  __I also missed more actual baselines that have been employed on the task or similar ones.__
>
> Since this is a new online LLM detection setting, we are not aware of any methods that have been applied to this task. Furthermore, as our work concerns the sequential hypothesis testing problem, we consider using permutation tests as our baseline, which is regarded as a classical method for hypothesis testing. The details of the baseline can be found in Appendix G.

---

> ### Author Response · Authors · 2024-11-18
> **Authors' response  (2/2)**
>
> * __Although the authors provided a quite detailed description of the experimental setup, it is not clear to what extent it is reproducible. We do need both validation and characterization of the dataset used in the experiments.__
>
> Thanks for raising the concern. We strive to ensure that our research can be thoroughly replicated by others. We have included comprehensive documents of the datasets and codes in our supplementary materials.
>
> &nbsp;
>
> * __I miss a contextualization of the state of the art of LLM-generated text detection – are those algorithms effective?__
>
> More historical context on related works in *offline* LLM-generated text detection is provided in Appendix A, and there are $10$ score functions (or the detectors) in the literature described in detail in Appendix B.
>
> &nbsp;
>
> * __It seems that an important characteristic of the problem tackled by the authors is that the paper tries to classify sources instead of individual documents – hence, the signal and confidence is dramatically higher than when you have to judge a single document. Consider highlighting this and take a look at https://arxiv.org/html/2406.07016v1.__
>
> We would like to emphasize that the online testing scenario in our paper involves maintaining an anytime-valid level test while controlling the time to reject the null hypothesis when the alternative is true (i.e., controlling the time to correctly  identify the source as an LLM), where a sample from the unknown source arrives sequentially. This presents a different problem compared to existing works on LLM-text detection, which typically frame their task as a standard classification problem. Hence, we believe that the problems may not be directly comparable. We would like to note that the problem and the solution we propose are particularly useful when one seeks substantial savings in both data collection and time without compromising the reliability of their statistical testing. These desiderata might be difficult to achieve with approaches based on collecting data in batches (or offline).
>
> Thank you for the reference, which is helpful in expanding our review of the relevant literature.
>
> &nbsp;
>
> [1] Guangsheng Bao, Yanbin Zhao, Zhiyang Teng, Linyi Yang, and Yue Zhang. Fast-detectgpt: Efficient zero-shot detection of machine-generated text via conditional probability curvature. arXiv preprint arXiv:2310.05130, 2023.
>
> [2] Eric Mitchell, Yoonho Lee, Alexander Khazatsky, Christopher D Manning, and Chelsea Finn. Detectgpt: Zero-shot machine-generated text detection using probability curvature. In International Conference on Machine Learning, pp. 24950–24962. PMLR, 2023.
>
> [3] Jinyan Su, Terry Yue Zhuo, Di Wang, and Preslav Nakov. Detectllm: Leveraging log rank information for zero-shot detection of machine-generated text. arXiv preprint arXiv:2306.05540, 2023.

---

> ### Author Response · Authors · 2024-11-25
>
> Dear Reviewer p6Pn,
>
> We appreciate your time in reviewing our manuscript and offering good suggestions. We have addressed the initial concerns you raised and submitted our responses a week ago. If there are any further issues that require clarification, we are more than willing to continue our discussion.

---

### Official Review · Reviewer_XdTt · 2024-11-08

**Soundness:** 4
**Presentation:** 3
**Contribution:** 3
**Rating:** 6
**Confidence:** 3

**Summary:**

The paper addresses the challenge of detecting large language model (LLM)-generated text in an online, streaming context. Traditional methods focus on offline detection, but the authors propose a sequential hypothesis testing algorithm that enables real-time detection with statistical guarantees, such as controlling the false positive rate and expected detection time. This method leverages online optimization and betting strategies to discern between human and LLM-generated content across platforms like social media and news websites.

**Strengths:**

S1. The problem of detecting LLM-generated text in an online, streaming context is important.

S2. The problem statement and the rationale behind the method are clear.

S3. The algorithm ensures a controlled false positive rate and has bounded detection time, which is crucial for reliability.

S4. The evaluation is very comprehensive. The paper tests the method with multiple score functions, datasets, and LLM configurations.

**Weaknesses:**

1. The method relies heavily on existing score functions from offline detection methods, potentially limiting its adaptability to new types of LLM-generated content. Certain score functions perform significantly better than others, indicating that the algorithm's efficacy may vary depending on the selected score function and dataset.

2. The experiments primarily focus on LLM-generated news about the Olympics and a pre-collected set of human-written news articles from XSum. This setup is limited in scope and may not reflect the diversity of content types that an LLM could generate. Would additional real-world validations in social media or low-quality content improve confidence in the algorithm’s effectiveness across various online platforms? (This comment doesn't lower my rating for this article, but rather my expectations for follow-up work)

3. The algorithm requires estimations for parameters like $\epsilon$ and $d$ before the main process begins. This estimation is based on initial samples, which introduces the risk of inaccurate settings, especially if the initial sample size is small. Can further optimization in estimating parameters improve consistency across different datasets and score functions?

**Questions:**

Please see Weaknesses.

---

> ### Author Response · Authors · 2024-11-18
>
> Thanks for your positive feedback and good suggestions. We appreciate the opportunity to discuss our work further.
>
> &nbsp;
>
> * __The method relies heavily on existing score functions from offline detection methods, potentially limiting its adaptability to new types of LLM-generated content. Certain score functions perform significantly better than others, indicating that the algorithm's efficacy may vary depending on the selected score function and dataset.__
>
> As we have discussed in the last section, although the choice of detector can influence the algorithm's performance, our experiment results show that our method equipped with most score functions performs well, which maintains the low type-1 error under the null hypothesis and rejects $H_0$ correctly when the alternative hypothesis is true.
> That is, our approach has strong statistical guarantees. Our method is designed to control the False Positive Rate (FPR) while being able to quickly detect LLM-generated text.
> In practice, although the performance of our method varies with different score functions, the experimental results consistently show that our method works as expected and aligns with the theoretical results.
>
> &nbsp;
>
> * __The experiments primarily focus on LLM-generated news about the Olympics and a pre-collected set of human-written news articles from XSum. This setup is limited in scope and may not reflect the diversity of content types that an LLM could generate. Would additional real-world validations in social media or low-quality content improve confidence in the algorithm’s effectiveness across various online platforms? (This comment doesn't lower my rating for this article, but rather my expectations for follow-up work)__
>
> We would like to clarify that we also have more experimental results
> in Appendix H for additional datasets such as WritingPrompts and PubMed. Moreover, we have tested scenarios where the target sequence of texts is of a different domain/topic from that of the prepared human-written texts (e.g., Figure 12). We acknowledge the value of extending our validations to include more content types on social media and other platforms and will consider exploring that in the future research.
>
> &nbsp;
>
> * __The algorithm requires estimations for parameters like $\epsilon$ and $d$ before the main process begins. This estimation is based on initial samples, which introduces the risk of inaccurate settings, especially if the initial sample size is small. Can further optimization in estimating parameters improve consistency across different datasets and score functions?__
>
> We believe that a reliable estimation of the parameters for our algorithm can enhance its performance. To support this conjecture, we include a new experiment in the updated version (refer to Figure 5~(b) in page 10), where we double the number of rounds in the initial stage of estimating the parameter $d_t$ and $\epsilon$ (from $10$ to $20$). The results support that using a larger sample size for estimation could potentially yield better outcomes.
>
> We also emphasize that if prior knowledge is available, estimating the required parameters is unnecessary. For instance, if one already knows the output range of an off-the-shelf score function and the discrepancy in scores written by different humans, the estimation is not required (this corresponds to the oracle scenario in the experiments), which could be a realistic scenario.

---

> > ### Comment · Reviewer_XdTt · 2024-11-20
> > **Official Comment**
> >
> > I appreciate the author's response and I have read all the reviewers' comments. I tend to keep the current score.

---

> > > ### Author Response · Authors · 2024-11-22
> > >
> > > We thank the reviewer again for your positive and constructive feedback!

---

### Official Review · Reviewer_y933 · 2024-11-08

**Soundness:** 3
**Presentation:** 4
**Contribution:** 3
**Rating:** 8
**Confidence:** 3

**Summary:**

In this paper, the authors present an online sequential hypothesis test for flagging LLM texts. At each time step, the algorithm updates an accumulated reward.

The reward increases whenever an algorithm scores higher for an LLM text, indicating that we have evidence to reject the null. When this is not the case, the authors control the reward so that it is never negative via an Online Newton algorithm. This is simple because the score functions are bounded (commonly between 0 and 1).

Proofs that the algorithm works are adapted from the literature on sequential hypothesis tests. I point out that this is now my area of expertise, but they seem reasonable. Overall, being one of the first approaches of this kind, the algorithm naturally achieves good results (the baseline is a quite simple approach from 10 years ago).

**Strengths:**

- Based on the authors' text, this is the first (if not the first) work to pose this problem. The algorithm is relatively easy to understand, although the presentation could improve slightly (see weaknesses) the first.

- The authors experiment with several different LLM detectors, showing how their method is easily adaptable to several LLM detection models.

- An extensive discussion of the models used and details of the method are presented in the appendixes. I did not double-check these; I only skimmed them, but they are welcomed.

**Weaknesses:**

Please provide an intuition for Line 11 earlier in the text. This is a crucial step of the algorithm, and I searched for the term "dt" on the pdf to better understand it. Ultimately, it is pretty intuitive since the score function is bounded. Nevertheless, with the explanation being brief and after the algorithm, it took me some time to understand it.

The paper does not discuss in detail why this approach is necessary, which is a significant weakness. Online testing may be employed, but is it the correct approach? For instance, how can I use this when evaluating student essays? I have only one essay per student. Maybe we could accumulate evidence that the account is a bot on a social media page. However, has this not already been done in batches?

The point above is not a weakness of the method but of the use case. Please clarify why this method is needed. Some examples of online forums are present, but were these attacks sequential? Were they in batches?

**Questions:**

See weaknesses. In particular, please answer the questions on the use case.

---

> ### Author Response · Authors · 2024-11-18
>
> We first would like to thank the reviewer for the positive feedback and useful comments.
>
> &nbsp;
>
> * __Intuition for Line 11 earlier in the text__
>
> Thanks for the suggestion.  We have updated accordingly in the new version for the clarity: $d_t$ is the upper bound of the absolute value of the difference between the score
> $\phi(x_t)$ and $\phi(y_t)$ at $t$, i.e., $d_t\geq | \phi(x_t) - \phi(y_t) |$. When updating to $t+1$, the information $d_{t+1}$ is used to determine the decision space of the update to ensure the wealth $\{W_t\}$ is a non-negative martingale, which is critical to control the type-1 error under the null hypothesis.
>
> &nbsp;
>
> * __Online testing may be employed, but is it the correct approach? For instance, how can I use this when evaluating student essays? I have only one essay per student. Maybe we could accumulate evidence that the account is a bot on a social media page. However, has this not already been done in batches?__
>
> Thanks for the great suggestion on our writing. We have also updated accordingly. We think online testing (i.e., maintaining an any-time valid level test while controlling the time to reject the null when the alternative is true as in our paper)
> is a correct approach or particularly useful when one seeks substantial savings in both data collection and time without compromising the reliability of their statistical testing.
> These desiderata might be elusive for approaches that based on collecting data in batch (or offline) to achieve.
> As for the scenario of one essay per student, we think that one may use any offline detector to classify the essay.
>
> One of the motivated scenarios that we mentioned in the paper is
> online bot detection in social media, and we believe a notable difference between our work and those detecting bots on social media pages (e.g., [1] and references therein) is that our work concerns a novel online setting with the goal of obtaining an any-time valid test with statistical guarantees that remains valid at all stopping times. Specifically, to the best of our knowledge, existing works (e.g., [1-4]) for bot detection don't provide strong theoretical guarantees, and they require training on extensive labeled datasets offline/beforehand.
>
> &nbsp;
>
> * __Some examples of online forums are present, but were these attacks sequential? Were they in batches?__
>
> Comments and posts are typically generated sequentially in social media. There are some papers/manuscripts (e.g., [5-7]) that provide some examples where the machine-generated texts were produced in response to ongoing discussions, which might indicate that the attack is in a sequential fashion. On the other hand, even if comments generated by LLMs are created in large quantities all at once, they are likely to be posted or displayed one after another (sequentially) on online platforms. Because of this sequential nature of their appearance, we believe that the online testing we concern is a relevant/practical scenario. We thank again for the reviewer's constructive feedback, which helps improve our paper.
>
> &nbsp;
>
> [1] Emilio Ferrara. Social bot detection in the age of chatgpt: Challenges and opportunities. First
> Monday, 2023
>
> [2] Alessandro Flammini, and Filippo Menczer.
> Botornot: A system to evaluate social bots. In Proceedings of the 25th international conference
> companion on world wide web, pp. 273–274, 2016.
>
> [3] Iacopo Pozzana and Emilio Ferrara. Measuring bot and human behavioral dynamics. Frontiers in Physics, 8:125, 2020.
>
> [4] Onur Varol, Emilio Ferrara, Clayton Davis, Filippo Menczer, and Alessandro Flammini. Online human-bot interactions: Detection, estimation, and characterization. In Proceedings of the inter-national AAAI conference on web and social media, volume 11, pp. 280–289, 2017.
>
> [5] A Selyukh. Fcc repeals ‘net neutrality’rules for internet providers. NPR (accessed 13 October 2020), 2017.
>
> [6] Max Weiss. Deepfake bot submissions to federal public comment websites cannot be distinguished
> from human submissions. Technology Science, 2019121801, 2019.
>
> [7] Y. Kilcher. This is the worst ai ever, June 2022.

---

> > ### Comment · Reviewer_y933 · 2024-11-20
> > **Thank you for your response**
> >
> > I thank the authors for their response. Given that my score is already positive, I shall keep my rating.

---

> > > ### Author Response · Authors · 2024-11-22
> > >
> > > We sincerely appreciate the reviewer's positive feedback. We deeply value your constructive suggestions!

---

### Author Response · Authors · 2024-11-25
**Summary of Revisions and Responses to Reviewers**

We thank the reviewers for taking their time to read our paper and providing constructive feedback. We would like to take the opportunity to summarize the updates that we have made during this rebuttal period.

A couple of reviewers commented that our experiments were conducted using a single dataset and/or using a new dataset rather than existing ones. We have clarified that additional experimental results for online testing are provided in Appendix H of our paper, where samples of LLM-generated texts and human-written texts (from datasets such as XSum, WritingPrompts, and PubMed) are from existing sources in the literature [1], with the LLM-generated texts produced by models such as GPT-3, ChatGPT, and GPT-4. We created a new dataset of Olympic 2024 news because we would like to validate the effectiveness of our approach in a more realistic scenario. Our codes and the new dataset have been uploaded for the reader to reproduce the results. As suggested by Reviewer p6Pn, we also have described the process of constructing the dataset and has included it in the updated version.

Second, to address the concern/question by Reviewer XdTt about specifying the parameters in our algorithm, we have added a new experimental result, where we double the number of rounds in the initial stage of estimating the parameter $d_t$ and $\epsilon$ (from $10$ to $20$). The results (Figure 5~(b) on page 10) indicate that using a larger sample size for estimation could potentially yield better outcomes.  We also emphasize that if prior knowledge is available, estimating the required parameters is unnecessary. For instance, if one already knows the output range of an off-the-shelf score function and the discrepancy in scores written by different humans, the estimation is not required. This corresponds to the oracle scenario in the experiments, which could be realistic in certain cases.
Additionally, we have included two more new experiments (Figure 6 on page 10), as suggested by Reviewer NdvJ, where we consider (1) tackling the case when the sequence of texts are generated by a few LLMs instead of a single one, (2) when the sequence contains a mix of human-written texts and LLM-generated texts, rather than exclusively one or the other. The new experimental results demonstrate our method effectively handles these scenarios as well.

In terms of writing/presentation, we have updated our paper accordingly based on the reviewers' feedback. As suggested by Reviewer y933 and Reviewer p6Pn, we have updated and highlighted in the paper the the online testing that our paper concerns (i.e.,
texts are observed sequentially, and one has to maintain an any-time valid level test while controlling the time to declare the source are LLMs when they are) can be particularly useful when one seeks substantial savings in both data collection and time without compromising the reliability of their statistical testing, which might be challenging for approaches that based on collecting data in batch (or offline) to achieve. To the best of our knowledge, this is a new scenario, whereas related works on offline detection focus on classifying a given set of texts without emphasizing such kind of statistical guarantees and may not be directly applicable to online detection. The tasks (i.e., online vs. offline) are fundamentally different, and therefore the corresponding methods might not be directly comparable. Specifically, we highlight in our response to Reviewer 87z9 that many state-of-the-art offline detection methods aim to determine whether a text is from an LLM by comparing its evaluated text score to a pre-specified threshold. A straightforward way to adapt an offline detector to our scenario is to declare the source as LLM (and stop) at the point where the detector predicts that a sample is written by an LLM. However, this approach will have a false positive rate of 1 when the number of rounds becomes sufficiently large, unless the offline detector never makes an error in recognizing human texts.

[1] Guangsheng Bao, Yanbin Zhao, Zhiyang Teng, Linyi Yang, and Yue Zhang. Fast-detectgpt: Efficient zero-shot detection of machine-generated text via conditional probability curvature. arXiv preprint arXiv:2310.05130, 2023.

---

### Meta-Review · Area_Chair_K4HS · 2024-12-17

**Metareview:**

The paper tackles the problem of detecting whether a text is either written by a large language model or a human in an online setting. To this end, it reduces the problem to a sequential hypothesis testing problem and adapts state of the art techniques. In terms of strengths, the reviewers appreciate the broad problem of detecting text generated by LLMs is certainly timely and important and the theoretical guarantees of the proposed methodology. In terms of weaknesses, the reviewers highlight insufficient motivation and contextualization with the state of the art and several issues regarding the experimentents. In terms of overall recommendation, the reviewers differ significantly and, as a result, I did read the manuscript to better decide about my recommendation. Based on the reviews and my own reading, my recommendation is to reject the paper. The two main reasons are:

(i) the actual application is not very well motivated, as highlighted by several reviewers, and the rebuttal by the authors did not persuade me otherwise. In that context, the authors do not sufficiently engage with related work on detection of LLM generated text based on, e.g., watermarking nor with related work on sequential hypothesis testing.

(ii) both in the main and the supplementary, it is unclear/difficult to realize to what extent the authors claim the methodology/technique to be a contribution or to be a straightforward adaptation of previous work on sequential hypothesis testing. In that context, one of the main differences with previous work seems to be the assumption on the bound for g_t, however, the authors fail to explain why the assumption from previous work does not hold in their case and instead needs to be related both in their manuscript and the rebuttal. Further, if the authors indeed claim the methodology advances the state of the art in sequential hypothesis testing, then, the framing of the manuscript is misleading.

**Additional Comments On Reviewer Discussion:**

The authors did provide a rebuttal to address the concerns raised by the reviewers and some of the reviewers followed up but they did not change their overall recommendation. In my view, the rebuttal did not completely clear out the concerns regarding motivation, experimental setup, and contextualization with the state of the art.

---

### Decision · Program_Chairs · 2025-01-22

Reject